A novel chaperone-effector-immunity system identified in uropathogenic Escherichia coli UMN026

Casiano González América 1
Pacheco Villanueva Arantxa 1
Castro-Alarcón Natividad 1
Méndez Julio 2
Oropeza Ricardo 2 ricardo.oropeza@ibt.unam.mx
http://orcid.org/0000-0002-0157-5696 Martínez-Santos Verónica I. 3 iranzu23@gmail.com
1 Microbiology Research Laboratory, Faculty of Chemical Biological Sciences, Universidad Autónoma de Guerrero , Chilpancingo, Guerrero , Mexico
2 Department of Molecular Microbiology, Institute of Biotechnology, Universidad Nacional Autónoma de México , Cuernavaca, Morelos , Mexico
3 CONAHCYT - Universidad Autonoma de Guerrero , Chilpancingo, Guerrero , Mexico
Mora-Montes Héctor
Electronic publication date: 2024 May 20
Publication date: 2024
Volume: 12
Electronic Location ID: e17336
Received 2023 Dec 18; Accepted 2024 Apr 15
Copyright: © 2024 Casiano González et al.
Copyright year: 2024
Copyright holder: Casiano González et al.
License: This is an open access article distributed under the terms of the Creative Commons Attribution License, which permits unrestricted use, distribution, reproduction and adaptation in any medium and for any purpose provided that it is properly attributed. For attribution, the original author(s), title, publication source (PeerJ) and either DOI or URL of the article must be cited.
License URL: https://creativecommons.org/licenses/by/4.0/

Keywords: UTI symptoms, Chaperone-effector-immunity system, Bacterial competence, Urinary tract infections, Uropathogenic Escherichia coli, Type 6 secretion system

Funding: PAPIIT/DGAPA Projects IN212619 and IN215222 CONAHCyT 764584 and 1006331 This work was supported by PAPIIT/DGAPA projects IN212619 and IN215222 to Ricardo Oropeza. América Casiano González and Arantxa Pacheco Villanueva were supported by scholarships from CONAHCyT (764584 and 1006331, respectively). There was no additional external funding received for this study. The funders had no role in study design, data collection and analysis, decision to publish, or preparation of the manuscript.

==============================
Background

Urinary tract infections (UTIs) are very common worldwide. According to their symptomatology, these infections are classified as pyelonephritis, cystitis, or asymptomatic bacteriuria (AB). Approximately 75–95% of UTIs are caused by uropathogenic Escherichia coli (UPEC), which is an extraintestinal bacterium that possesses virulence factors for bacterial adherence and invasion in the urinary tract. In addition, UPEC possesses type 6 secretion systems (T6SS) as virulence mechanisms that can participate in bacterial competition and in bacterial pathogenicity. UPEC UMN026 carries three genes, namely, ECUMN_0231, ECUMN_0232, and ECUMN_0233, which encode three uncharacterized proteins related to the T6SS that are conserved in strains from phylogroups B2 and D and have been proposed as biomarkers of UTIs.

Aim

To analyze the frequency of the ECUMN_0231, ECUMN_0232, ECUMN_0233, and vgrG genes in UTI isolates, as well as their expression in Luria Bertani (LB) medium and urine; to determine whether these genes are related to UTI symptoms or bacterial competence and to identify functional domains on the putative proteins.

Methods

The frequency of the ECUMN and vgrG genes in 99 clinical isolates from UPEC was determined by endpoint PCR. The relationship between gene presence and UTI symptomatology was determined using the chi2 test, with p < 0.05 considered to indicate statistical significance. The expression of the three ECUMN genes and vgrG was analyzed by RT-PCR. The antibacterial activity of strain UMN026 was determined by bacterial competence assays. The identification of functional domains and the docking were performed using bioinformatic tools.

Results

The ECUMN genes are conserved in 33.3% of clinical isolates from patients with symptomatic and asymptomatic UTIs and have no relationship with UTI symptomatology. Of the ECUMN+ isolates, only five (15.15%, 5/33) had the three ECUMN and vgrG genes. These genes were expressed in LB broth and urine in UPEC UMN026 but not in all the clinical isolates. Strain UMN026 had antibacterial activity against UPEC clinical isolate 4014 (ECUMN−) and E. faecalis but not against isolate 4012 (ECUMN+). Bioinformatics analysis suggested that the ECUMN genes encode a chaperone/effector/immunity system.

Conclusions

The ECUMN genes are conserved in clinical isolates from symptomatic and asymptomatic patients and are not related to UTI symptoms. However, these genes encode a putative chaperone/effector/immunity system that seems to be involved in the antibacterial activity of strain UMN026.

Introduction

Uropathogenic Escherichia coli (UPEC) is a Gram-negative bacterium that colonizes the urinary tract and is the main etiological agent of urinary tract infections (UTIs), causing approximately 80% of cases (Terlizzi, Gribaudo & Maffei, 2017). These infections are classified according to their symptomatology as symptomatic UTIs (cystitis and pyelonephritis) or asymptomatic bacteriuria (AB) (Millner & Becknell, 2019). Cystitis is characterized by dysuria, frequent evacuation of small volumes, and urinary urgency (Colgan & Williams, 2011), while pyelonephritis is characterized by fever, chills, flank pain, nausea and vomiting (Castaigne, Georges & Jouret, 2022).

UPEC strains utilize several virulence factors, including adhesins (e.g., type 1, P, and curli fimbriae) and toxins (LPS, a-hemolysin, CFN1, among others), to establish infection (Terlizzi, Gribaudo & Maffei, 2017). Recently, it has been shown that UPEC possesses a secretion system (SS) known as the type VI secretion system (T6SS). Bacterial T6SSs are evolutionarily related to the tails of contractile tailed bacteriophages, thus their structure and function are similar (Journet & Cascales, 2016). These systems are formed by three complexes: the membrane complex, the base plate complex, and the syringe. The latter is formed by an inner tube composed of the Hcp protein, and has a tip formed by a trimer of the VgrG protein. The Hcp and VgrG proteins are the hallmark of the T6SS (Cascales & Cambillau, 2012). These systems have been classified into three phylogenetic groups according to gene organization and homology: T6SS-1, -2, and -3. T6SS-2 is usually associated with bacterial virulence due to its role in colonization, survival, and invasion, while T6SS-1 and T6SS-3 are not involved directly in virulence, but rather are involved in antibacterial activity or bacterial competition (Journet & Cascales, 2016). UPEC strains have been shown to possess mainly T6SS-1 (like strain CFT073) and T6SS-2 (like strain UMN026) (Journet & Cascales, 2016; Nielsen et al., 2017). Initially, the role of T6SSs in bacterial virulence was associated with host damage; however, their main role is related to bacterial competition due to the secretion of effector proteins, some of which are effector-immunity (E-I) pairs (Dong et al., 2013; Hood et al., 2010; Navarro-Garcia et al., 2019). Effector proteins are antibacterial toxins that attack bacterial targets, such as bacterial cell walls (muramidases and glycoside hydrolases), nucleic acids (nucleases), and membrane structures (lipases). These toxins are neutralized by cognate immunity proteins that neutralize them by direct binding, thus preventing auto or sibling intoxication. Genes encoding E-I pairs are always adjacent (Koskiniemi et al., 2013; Ma et al., 2014; Russell et al., 2013). Since effector proteins target bacterial structures, they mediate bacterial competition. For example, the Pseudomonas protegens effector protein Tge2, a peptidoglycan glycoside hydrolase effector that is inactivated by its cognate immunity protein Tgi2, is involved in bacterial competition against P. putida (Whitney et al., 2013).

Nielsen et al. (2017) found three family genes that were overrepresented in UTI isolates in comparison to fecal isolates. These genes encode putative T6SS proteins, are located in a T6SS cluster, and were found only in isolates belonging to phylogroups B2 and D. Therefore, the authors suggested that these genes could be predictor biomarkers for UTIs. These genes are annotated as ECUMN_0231, ECUMN_0232, and ECUMN_0233 in the genome of UPEC UMN026, and the T6SS cluster is type 2 (Journet & Cascales, 2016; Nielsen et al., 2017). The aim of this work was to analyze the frequency of the ECUMN genes, as well as the frequency of the vgrG gene (a core component of the T6SS); to analyze the expression of these genes; to relate the presence of these genes to UTI symptomatology; to identify functional domains in the putative proteins and predict protein-protein interactions.

Materials and Methods

Bacterial strains and growth conditions

A total of 99 previously isolated UPEC clinical isolates were used (Martinez-Santos et al., 2021). The reference strains and clinical isolates used for RT-PCR and antibacterial activity assays are described in Table 1. Bacteria were grown at 37 °C under static or shaking conditions in LB broth (5 g/L NaCl, 5 g/L yeast extract, 10 g/L casein peptone) (Dibico, Mexico) or filtered urine. Mid-stream urine was collected from healthy volunteers by the mid-stream-clean-catch technique (MSCC) (Maher, Brown & Gatewood, 2017), centrifuged at 4,000 rpm at 4 °C for 10 min, and filtered with 0.22 µm sterile Millex-GP syringe filters (Merck Millipore, Germany). When necessary, the medium was supplemented with kanamycin (Km, 40 μg/mL).

Table 1 Reference strains and clinical isolates used in antibacterial activity assays.

Strain	Characteristics	Reference	
UPEC UMN026 (ATCC BAA-1161)	Clonal group A urine isolate from a female with acute cystitis. ACSSuTTp resistant. ECUMN_0231+, ECUMN_0232+, ECUMN_0233+. KmS.	American type culture collection	
UPEC 4039	Clinical isolate from a female with AB. Phylogroup D.	Martinez-Santos et al. (2021)	
UPEC 4073	Clinical isolate from a female with pyelonephritis. Phylogroup D.	
UPEC 4079	Clinical isolate from a female with pyelonephritis. Phylogroup D.	
UPEC 4087	Clinical isolate from a female with pyelonephritis. Phylogroup Clade I.	
UPEC 4012	Clinical isolate from a female with AB used as prey in bacterial competition assays. Phylogroup A. ECUMN_0231+, ECUMN_0232+, ECUMN_0233+. KmR.	
UPEC 4014	Clinical isolate from a female with AB used as prey in bacterial competition assays. Phylogroup B2. ECUMN_0231−, ECUMN_0232−, ECUMN_0233−. KmR.	
UPEC 4018	Clinical isolate from a female with AB used as negative control in bacterial competition assays. Phylogroup A. ECUMN_0231−, ECUMN_0232−, ECUMN_0233−. KmS.	
Staphylococcus aureus ATCC 29213	Quality control strain isolated from wound used as prey in bacterial competition assays. KmR.	American type culture collection	
Enterococcus faecalis ATCC 29212	Quality control strain isolated from urine used as prey in bacterial competition assays. KmR.	
Klebsiella pneumoniae K6 ATCC 700603	Quality control strain isolated from urine of a hospitalized patient used as prey in bacterial competition assays. KmR.	
Note:

ACSSuTTp: ampicillin, chloramphenicol, streptomycin, sulfonamides, tetracycline, trimethoprim (Lescat et al., 2009). KmS: kanamycin sensitive, KmR: kanamycin resistant.

Oligonucleotides

The oligonucleotides used in this work were synthesized by the Oligonucleotide Synthesis Facility at Instituto de Biotecnologia (UNAM, Cuernavaca, Mexico) and are listed in Table 2.

Table 2 Oligonucleotides used in this work.

Oligonucleotide	Sequence (5′-3′)	Tm (°C)	Amplicon size (bp)	Reference	
VgrG_qPCRF	TGCAAGTCGTTGTGCTGAAC	59.6	105	This study	
VgrG_qPCRR	TCACTTTCCGGTCATTTGC	58.8	This study	
0231_qPCRF	GCTGGATTTTCTGAACGAGGTG	59.8	129	This study	
0231_qPCRR	ACACTCAGCCATGCTTCAAC	58.8	This study	
0232_qPCRF	TAAAAGCCAGGGACGCAATG	58.8	127	This study	
0232_qPCRR	ACGCCAAAATCGACAGTCAG	58.9	This study	
0233_qPCRF	ATGATGCTGATGGCGAATGG	58.8	79	This study	
0233_qPCRR	TGGCAGGCTTGTCAGAAAAC	59.0	This study	

DNA extraction

Total DNA was obtained from the clinical isolates by thermal shock as described previously (Hernandez-Vergara et al., 2016). Briefly, 5–6 colonies were resuspended in 100 μL of sterile water, frozen at −70 °C for 30 min, and boiled at 96 °C for 10 min. The tubes were centrifuged at 13,200 rpm for 10 min, the supernatant was recovered, and the DNA was quantified on a NanoDrop 2000 (Thermo Scientific, Waltham, MA, USA).

Amplification of ECUMN genes

The frequency of the ECUMN genes _0231, _0232, and _0233, as well as the vgrG gene, was determined by PCR using specific primers (Table 2). The reactions were performed in a final volume of 12.5 μL, containing: 1X buffer, 25 mM MgCl2, 1 mM dNTPs, 10 μM of each oligonucleotide, 150 ng of DNA, and 1.25 U of Taq DNA polymerase (Thermo Scientific, Waltham, MA, USA). The following conditions were used: 1 cycle at 95 °C for 5 min; 30 cycles at 95 °C for 1 min, 55 °C for 1 min, 72 °C for 15 s; and 1 cycle at 72 °C for 10 min for vgrG, ECUMN_0231, and ECUMN_0233. For ECUMN_0232, the following conditions were used: 1 cycle at 95 °C for 5 min; 30 cycles at 95 °C for 1 min, 58 °C for 1 min, 72 °C for 15 s; and 1 cycle at 72 °C for 10 min. The PCR products were analyzed on 2% agarose gels stained with ethidium bromide.

RNA purification and retro-transcription

Three mL of LB broth were inoculated with UPEC strain UMN026 and incubated overnight at 37 °C. Then, 25 mL of LB broth or urine were inoculated with 250 μL of the preinoculum and incubated at 37 °C with or without shaking for 4 (LB) or 15 h (urine). Bacterial cells were collected by centrifugation at 4,000 rpm for 15 min at 4 °C, and the pellets were stored overnight at −80 °C. Total RNA was purified with TRIzol. RNA integrity was verified in 1% agarose gels with 6% bleach (Aranda, LaJoie & Jorcyk, 2012). Purified RNA was treated with DNAse I (Thermo Scientific, Waltham, MA, USA) at 37 °C for 30 min. The reaction was performed in a final volume of 15 μL, containing 10 μL of RNA, 1.5 μL of 10X buffer, and 1.5 μL of DNAse I (1 U/mL). The enzyme was inactivated by adding 1 μL of 50 mM EDTA and incubating at 65 °C for 10 min. cDNA was synthetized using 1 μg of total RNA treated with DNAse I and the Verso cDNA synthesis kit (Thermo Scientific, Waltham, MA, USA), according to the manufacturer’s instructions.

Expression of the ECUMN genes

The expression of the ECUMN genes was determined by PCR using specific primers (Table 2). The reactions were performed in a final volume of 12.5 μL, containing: 1X buffer, 25 mM MgCl2, 1 mM dNTPs, 10 μM of each oligonucleotide, 150 ng/μL cDNA, and 1.25 U Taq DNA polymerase (Thermo Scientific, Waltham, MA, USA). The following conditions were used: 1 cycle at 95 °C for 5 min; 30 cycles at 95 °C for 1 min, 55 °C for 1 min, and 72 °C for 15 s; and 1 cycle at 72 °C for 10 min for vgrG, ECUMN_0231, and ECUMN_0233. For the expression of ECUMN_0232, the following conditions were used: 1 cycle at 95 °C for 5 min; 30 cycles at 95 °C for 1 min, 58 °C for 1 min, and 72 °C for 15 s; and 1 cycle at 72 °C for 10 min. The PCR products were analyzed in 2% agarose gels stained with ethidium bromide.

Antibacterial activity (bacterial competition assays)

Prey strains (4012, 4014, S. aureus, E. faecalis, and K. pneumoniae), as well as the attacking strains (UMN026 and 4018), were grown overnight at 37 °C in 3 mL of supplemented or unsupplemented LB broth or urine, respectively, with Km. One mL of each overnight culture was centrifuged at 13,000 rpm for 3 min, after which the pellets were washed with 1 mL of LB broth and then resuspended in LB broth, after which the OD600 was adjusted to 1. Cultures of attacking and prey strains were mixed at a 10:1 ratio in a total volume of 200 μL, spotted on LB plates, and incubated at 37 °C for 24 h. Bacterial growth was recovered in 2 mL of LB broth, the OD600 was normalized to 0.5, and 10 μL of serial dilutions (10−1 to 10−6) were spotted on LB plates supplemented with Km and incubated at 37 °C for 24 h. The colonies were photographed and recovered in 1 mL of LB broth, after which the OD600 was determined (Basler, Ho & Mekalanos, 2013; Fitzsimons et al., 2018; Ma et al., 2020). Antibacterial activity assays were performed in triplicate.

Statistical analysis

Statistical analysis was performed using the STATA software v. 13.1. The relationships between quantitative variables were determined using the chi2 test, and p < 0.05 was considered to indicate statistical significance.

Bioinformatic analysis

Identification of the possible functions of the three proteins ECUMN_0231 (GenBank: CAR11446.1), ECUMN_0232 (GenBank: CAR11447.1), and ECUMN_0233 (GenBank: CAR11448.1) was performed by available bioinformatic programs, using the online tools NCBI BLAST (https://blast.ncbi.nlm.nih.gov/) (Altschul et al., 1990), NCBI Conserved Domain Search (https://www.ncbi.nlm.nih.gov/Structure/cdd/wrpsb.cgi) (Lu et al., 2020; Marchler-Bauer et al., 2017; Wang et al., 2023), KEGG (https://www.genome.jp/kegg/) (Kanehisa & Goto, 2000), UNIPROT (https://www.uniprot.org) (UniProt, 2023), InterPro (https://www.ebi.ac.uk/interpro/) (Paysan-Lafosse et al., 2023), SignalP-6.0 (https://services.healthtech.dtu.dk/services/SignalP-6.0/) (Teufel et al., 2022), and Bastion6 (https://bastion6.erc.monash.edu/) (Wang et al., 2018). Protein sequences were used to identify functional domains and features. With the information collected, representative images of the conserved regions of each protein were constructed.

Prediction of protein-protein interactions (docking)

Modeling of protein interactions was performed using the software UCSF ChimeraX, which has an integrated protein structure prediction tool based on the AlphaFold2-multimer code that uses ColabFold.V1.5.5 (free access version) (Meng et al., 2023). The amino acid sequences of the ECUMN_0231, ECUMN_0232, and ECUMN_0233 proteins (accession numbers B7N896_ECOLU, B7N897_ECOLU and B7N898_ECOLU, respectively) were downloaded from UniProt. Possible protein-protein interactions (docking) were probed between proteins ECUMN_0231_vs_ECUMN_0232 and ECUMN_0232_vs_ECUMN_0233 entering the downloaded sequences separated by commas. ColabFold.V1.5.5 finished the run and automatically opened the best model obtained after several cycles. The “contacts” tool was used on UCSF ChimeraX to calculate the contacts with default options between proteins, showing the distances calculated by the software. Finally, the names and numbers of the residues involved in interactions were added.

Results

The ECUMN and vgrG genes are conserved in isolates from symptomatic and asymptomatic UTIs

As stated in the introduction, the ECUMN genes _0231, _0232, and _0233 were proposed as biomarkers of UTIs since they are conserved in strains isolated from patients with symptomatic UTIs (Nielsen et al., 2017). To corroborate whether these genes are conserved only in isolates from symptomatic patients, we determined the presence of these genes in clinical isolates obtained from patients with cystitis, pyelonephritis, and AB (Table 3). We found that of the 99 clinical isolates analyzed, 33 (33.3%) had at least one ECUMN gene, independent of the symptomatology of the patients, indicating that these genes are conserved not only in isolates from symptomatic UTI patients, but also in isolates from AB patients. Since the ECUMN genes are encoded in the T6SS cluster, we also determined the frequency of the vgrG gene. This gene was more frequent (48.5%) than the ECUMN genes, and it was also present in isolates from patients with and without symptoms.

Table 3 Symptomatology and presence of ECUMN genes.

Characteristics	Total
n = 99 (100%)	ECUMN+
n = 33 (33.3%)	ECUMN− n = 66 (66.7%)	p value	vgrG+
n = 48 (48.5%)	vgrG−
n = 51 (51.5%)	p value	
Symptomatology								
Symptomatic	45 (45.5)	11 (33.3)	34 (51.5)	0.08	20 (41.7)	25 (49)	0.46	
Asymptomatic	54 (54.5)	22 (66.7)	32 (48.5)		28 (58.3)	26 (51)		
Diagnostic								
AB	54 (54.5)	22 (66.7)	32 (48.5)	0.18	28 (58.3)	26 (51)	0.66	
Cystitis	25 (25.3)	5 (15.1)	20 (30.3)		12 (25)	13 (25.5)		
Pyelonephritis	20 (20.2)	6 (18.2)	14 (21.2)		8 (16.7)	12 (23.5)		
Note:

n-total number of clinical isolates. p value was calculated by the Chi2 test, p < 0.05 was considered statistically significant.

We also analyzed the individual conservation of these genes. As shown in Fig. 1, most of the isolates have only ECUMN_0231 (36.36%, 12/33) or ECUMN_0233 (30.3%, 10/33). Two isolates (6.06%) had only ECUMN_0232, or the combinations ECUMN_0232/_0233 and ECUMN_0231/_0233. Only five isolates (15.15%) had these three genes. Interestingly, no isolate contained the combination ECUMN_0231/_0232. Notably, of the five isolates with the three genes, three were from patients with pyelonephritis, and two were from patients with AB.

Figure 1 Frequency of ECUMN genes.

Pie chart showing the percentage of clinical isolates carrying ECUMN genes.

The ECUMN and vgrG genes are present in pathogenic and nonpathogenic phylogroups

It was previously reported that the ECUMN genes are conserved only in UPEC strains belonging to phylogroups B2 and D (Nielsen et al., 2017). To test this hypothesis, we also analyzed the frequency of the ECUMN genes according to phylogenetic group (Table 4). Our results showed that the ECUMN genes were more frequent in isolates belonging to phylogroup B2, followed by isolates of phylogroup D (both of which are considered pathogenic), although they were also found in isolates from commensal phylogroups, mainly phylogroup F. On the other hand, similar results were obtained for the vgrG gene, although in this case, the presence of the gene is not related to the phylogroup.

Table 4 Phylogroups and presence of ECUMN and vgrG genes.

Phylogroup	Total
n = 99
(100%)	ECUMN+
n = 33
(33.3%)	ECUMN−
n = 66
(66.7%)	p value	vgrG+
n = 48
(48.5%)	vgrG−
n = 51
(51.5%)	p value	
A	15 (15.2)	2 (6.06)	13 (19.7)	0.0048	6 (12.5)	9 (17.6)	0.533	
B1	11 (11.1)	2 (6.06)	9 (13.6)		5 (10.4)	6 (11.7)		
B2	51 (51.5)	15 (45.45)	36 (54.6)		22 (45.8)	29 (56.9)		
C	3 (3.03)	1 (3.03)	2 (3.03)		2 (4.2)	1 (2)		
D	11 (11.1)	8 (24.24)	3 (4.54)		7 (14.6)	4 (7.8)		
E	2 (2.01)	0 (0)	2 (3.03)		1 (2)	1 (2)		
F	3 (3.03)	3 (9.1)	0 (0)		3 (6.3)	0 (0)		
Clade	3 (3.03)	2 (6.06)	1 (1.5)		2 (4.2)	1 (2)		
Note:

n-total number of clinical isolates. p value was calculated by the Chi2 test, p < 0.05 was considered statistically significant.

The ECUMN and vgrG genes are expressed in LB and urine

Since the ECUMN genes are annotated as putative genes and there seems to be no relationship with symptoms, we analyzed their expression under laboratory (LB broth) and UTI-like conditions (urine) in the prototype strain UMN026. As shown in Fig. 2, all the genes were expressed in both LB broth and urine under shaking or static conditions. The same expression pattern was observed for the vgrG gene, indicating that the ECUMN genes are expressed under the same conditions as the T6SS.

Figure 2 Expression of ECUMN and vgrG genes in UPEC UMN026.

Expression of genes vgrG (A, B), ECUMN_0231 (C, D), ECUMN_0232 (E, F), and ECUMN_0233 (G, H) was determined from bacteria grown in LB broth and urine at 37 °C with shaking (Sha) and in static conditions (Sta). Genomic DNA was used as positive control (DNA) and RNA treated with DNAse I was used as negative control (RNA). MW, molecular weight marker.

To determine whether the clinical isolates also expressed these genes, we analyzed their expression in the clinical isolates that possessed all three genes. As shown in Figs. 3A and 3B, ECUMN_0231 was expressed in all the clinical isolates in LB, while in urine the clinical isolates 4087, 4012, 4039, and 4079, but not isolate 4073, expressed this gene (Figs. 3C and 3D).

Figure 3 Expression of ECUMN_0231.

Prototype strain UMN026 and clinical isolates 4012, 4039, 4073, 4079, and 4087 were grown in LB and urine at 37 °C in static conditions in LB broth (A and B) and urine (C and D). Lanes: MW, molecular weight marker; gDNA, genomic DNA used as positive control; RNA-RNA treated with DNAse I used as negative control; cDNA, complementary DNA used as template.

On the other hand, ECUMN_0232 was expressed in all the isolates in LB and urine (Fig. 4). A similar result was obtained with ECUMN_0233 (Fig. 5).

Figure 4 Expression of ECUMN_0232.

Prototype strain UMN026 and clinical isolates 4012, 4039, 4073, 4079, and 4087 were grown in LB and urine at 37 °C in static conditions in LB broth (A and B) and urine (C and D). Lanes: MW, molecular weight marker; gDNA, genomic DNA used as positive control; RNA, RNA treated with DNAse I used as negative control; cDNA, complementary DNA used as template.

Figure 5 Expression of ECUMN_0233.

Prototype strain UMN026 and clinical isolates 4012, 4039, 4073, 4079, and 4087 were grown in LB and urine at 37 °C in static conditions in LB broth (A and B) and urine (C and D). Lanes: MW, molecular weight marker; gDNA, genomic DNA used as positive control; RNA, RNA treated with DNAse I used as negative control; cDNA, complementary DNA used as template.

Finally, the analysis of vgrG expression showed that this gene is expressed in all isolates when grown in LB, while it is only expressed in UPEC UMN026 and the isolate 4087 when grown in urine (Fig. 6).

Figure 6 Expression of vgrG.

Prototype strain UMN026 and clinical isolates 4012, 4039 4073 4079 and 4087 were grown in LB and urine at 37 °C in static conditions in LB broth (A and B) and urine (C and D). Lanes: MW, molecular weight marker; gDNA, genomic DNA used as positive control; RNA, RNA treated with DNAse I used as negative control; cDNA, complementary DNA used as template.

UPEC UMN026 has inter- and intraspecific antibacterial activity

Our results showed that there was no relationship between the presence of the ECUMN genes and the symptomatology. However, T6SSs have also been shown to be involved in bacterial competence. To determine whether UPEC UMN026 is able to kill other bacteria, we performed bacterial competition assays against two UPEC clinical isolates (4012 and 4014), as well as against S. aureus, K. pneumoniae, and E. faecalis (Table 1). Our results showed that UPEC UMN026 did not have bacteriolytic activity against isolate 4012 (data not shown), but it has antibacterial activity against the clinical isolate 4014 when cultured both in LB and urine (Fig. 7A, Fig. S1A). When the assays were performed against other species, our results showed that UPEC UMN026 had no activity against K. pneumoniae or S. aureus (data not shown), but it had activity against E. faecalis (Fig. 7B, Fig. S1B). In this case, the effect was clearer in urine than in LB. These results indicate that UPEC UMN026 has intra- and interspecific antibacterial activity.

Figure 7 Antibacterial activity of UPEC UMN026.

Bacterial competence assays between UPEC UMN026 and clinical isolate 4014 (A) and E. faecalis (B) grown in LB broth and urine. Clinical isolate 4018 (Table 1) was used as negative control. The bars represent recovered bacteria from serial dilutions (10−1 to 10−6). ns, no significant; statistically significant p values are indicated.

ECUMN genes form a chaperone-effector-immunity system

To understand the function of the ECUMN genes, we performed a bioinformatic analysis. Blastp, UniProt, and KEGG analyses revealed that ECUMN_0231 has a DUF4123 domain (pfam13503) (residues 9–122) (Fig. 8B), which has approximately 120 residues and four conserved motifs. Genes encoding proteins containing the DUF4123 motif are commonly found downstream of vgrG or putative effector-encoding genes (Moriel et al., 2021; Unterweger et al., 2015). This domain has been shown to be present in T6SS effector chaperones (TECs) in several bacterial species. These chaperones have been shown to be necessary only to load the cognate effector onto the T6SS to be secreted (Liang et al., 2015; Unterweger et al., 2015) or to stabilize its cognate effector, maintain its levels and aid in its secretion (Bondage et al., 2016). The NCBI Conserved Domains tool classifies ECUMN_0231 as a DUF4123 domain-containing protein similar to the Vibrio cholerae accessory protein VasW that plays an accessory role in VasX-mediated bacterial killing. VasW is encoded immediately upstream of vasX, which encodes an effector protein, and is proposed to mediate the interaction between VasX and the T6SS tip protein PAAR, so VasX can be secreted (Miyata et al., 2013). Another DUF4123 domain-containing protein is Proteus mirabilis IdsC, which stabilizes the effector protein IdsD in subcellular clusters, maintaining its levels and aiding in its secretion through the T6SS (Zepeda-Rivera, Saak & Gibbs, 2018). Thus, this result suggested that ECUMN_0231 encodes a putative TEC (Fig. 8A). On the other hand, our in silico analysis using BLAST and Conserved Domains Search showed that ECUMN_0232 has no conserved domains; however the Bastion6 tool predicts that ECUMN_0232 is a T6S effector. Accordingly, its localization (Fig. 8A) suggests that it could be an effector protein since genes that encode T6S effectors are often located adjacent to or near vgrG (Koskiniemi et al., 2013; Ma et al., 2014; Russell et al., 2013), and downstream of genes that encode DUF4123 domain-containing proteins (Liang et al., 2015). This finding also suggested that ECUMN_0231 could be a chaperone of ECUMN_0232 since the DUF4123 domain containing proteins seem to be specific for the effector encoded immediately downstream (Bondage et al., 2016; Liang et al., 2015). Finally, UniProt and InterPro analyses revealed that ECUMN_0233 has two features, the first being a signal peptide on the first 18 residues of its N-terminal domain (MKAKHLISVILLSGVVMG; n-region residues 1–5, h-region residues 6–13, and c-region residues 14–18) (Fig. 8B). The second feature is a chain or a noncytoplasmic domain, which is described as the extent of a polypeptide chain in the mature protein following processing or proteolytic cleavage. Further analysis using SignalP-6.0 showed that ECUMN_0233 has a lipoprotein signal peptide (Sec/SPII) with a likelihood of 0.7699, and a cleavage site between positions 21 and 22 with a probability of 0.662976. Lipoprotein signal peptides are transported by the Sec translocon and cleaved by signal peptidase II (Lsp) (Teufel et al., 2022). This result suggested that this protein could be translocated to the periplasm through the Sec system. This finding suggested that ECUMN_0233 could be an immunity protein located in the periplasm or in the outer membrane of bacteria. Taken together these findings indicate that ECUMN_0231, ECUMN_0232, and ECUMN_0233 encode proteins that form a chaperone/effector/immunity system.

Figure 8 Organization and functional domains in ECUMN genes.

(A) Genomic organization of the ECUMN genes (gray arrows). The proposed function of each gene is indicated. TEC-T6S effector chaperone. The lengths of the intergenic regions are indicated. The vgrG and ECUMN_0231 genes overlap in three nucleotides. (B) Domains identified in ECUMN_0231 and ECUMN_0233 proteins (gray regions). DUF4123- domain of unknown function 4123, SP-signal peptide. The numbers indicate the number of residues and the positions containing the domains.

To support this hypothesis, we performed a prediction of protein-protein interactions. The docking between ECUMN_0231 and ECUMN_ 0232 predicted 64 true contacts. The predicted true contact distances ranged between 2.679 Å, the nearest interaction, and 4.132 Å, the farthest. The contact points in ECUMN_0231 were distributed along the protein, whereas the contact residues in ECUMN_0232 were located only at the N-terminal end (Fig. 9A). The predicted ECUMN_0231 residues involved in the interaction are L101, D227, D226, A225, P262, and Y261, while the residues in ECUMN_0232 are C12, S6, T7, A8, S39, and K40 (Fig. 9B). On the other hand, the docking between ECUMN_0232 and ECUMN_0233 showed 444 true contacts. The distances between the contacts ranged from 2.607 Å to 4.160Å. In contrast to the interaction with ECUMN_0231, the residues in ECUMN_0232 that interact with ECUMN_0233 were located on the C-terminal end, whereas for ECUMN_0233, the residues involved were distributed along the protein (Fig. 10).

Figure 9 Docking between ECUMN_0231 and ECUMN_0232.

(A) Protein-protein interaction modeled with AlphaFold2. ECUMN_0231 is shown in blue and ECUMN_0232 is shown in red. (B) Zoom in on the zone in the red square in (A). Residues of ECUMN_0231 involved in the interactions are LEU 101, ASP 227 and 226, ALA 225, PRO 262, and TYR 261. Contact residues in ECUMN_0232 are CYS 12, SER 6, THR 7, ALA 8, SER 39, and LYS 40. Contacts are indicated by yellow dashed lines.

Figure 10 Docking between ECUMN_0232 and ECUMN_0233.

(A) Protein-protein interaction modeled with AlphaFold2. ECUMN_0232 is shown in purple and ECUMN_0233 is shown in red. (B–D) Zoom in on the zones in the red, blue, and green squares in (A), respectively. Contacts are indicated by yellow dashed lines.

Discussion

UTIs are the most common infection in ambulatory and hospitalized patients, only after respiratory infections. The main etiological agent is UPEC, which utilizes several virulence factors to colonize the UT and cause infection. One of the virulence factors recently described is the T6SS, a molecular nanomachine that delivers effector proteins directly into the cytoplasm of bacterial or eukaryotic cells. This characteristic allows it to cause damage to the host, as well as to compete with other bacteria. Among the effector proteins secreted by this SS, are effector-immunity pairs, in which the effector is a toxin that attacks cellular components, and the immunity protein neutralizes the toxin by binding to it. UPEC UMN026 encodes a T6SS-2, and downstream of this cluster are the genes ECUMN_0231, ECUMN_0232, and ECUMN_0233. These genes were proposed to be predictor biomarkers of UTIs due to their overrepresentation in UTI isolates compared to fecal isolates (Nielsen et al., 2017). We analyzed the presence of these genes in 99 clinical isolates obtained from patients with symptomatic and asymptomatic UTIs and found that the genes are present in 33.33% of the isolates, a frequency similar to that found by Nielsen et al. (2017), who found these genes in approximately 40% of UTI isolates. We found that the genes were present not only in isolates from symptomatic but also from asymptomatic patients. We also found that these genes are more frequent in phylogroups B2, D, and F. The difference between our results and those reported by Nielsen et al. may be explained because we included patients with AB, as well as patients with ailments (type 2 diabetes mellitus, pregnancy, catheter, and chronic renal illness) (Martinez-Santos et al., 2021).

Some UPEC virulence factors, such as Cnf1 and HlyA, have been associated with UTI symptoms (Martinez-Santos et al., 2021); therefore, we analyzed whether there was any relationship between the expression of the ECUMN genes and disease symptoms. Interestingly, we found no relationship. This result was unexpected because T6SS-2 clusters are overrepresented in highly virulent pathogenic bacteria and have been shown to impact colonization, survival, and invasion (Journet & Cascales, 2016). Our result could be because the three ECUMN genes are not present in all the ECUMN+ isolates but are present in only five (15.15%) isolates, while most of the isolates have only one gene. This result is similar to that obtained by Nielsen et al. (2017) who found that of the 156 isolates analyzed (108 from feces and 48 from UTIs), 32 (20.5%) had the three genes. It has been shown that genes encoding E-I pairs are acquired by horizontal gene transfer in what has been called “effector modules” (Kirchberger et al., 2017; Unterweger et al., 2014), so it is possible that not all the genes are acquired together. Nielsen et al. (2017) found differences in the number of each gene present in their isolates, although they did not mention the combinations they found. Additionally, Journet & Cascales (2016) reported that the region containing these genes is not present or not identical in the strains analyzed. In the UPEC strains UTI89 and 536, the first gene of this region encodes a protein carrying a DUF2169 domain, and the second and third genes are smaller than ECUMN_0232 (1,440 bp vs 1,572 bp), and ECUMN_0233 (291 vs 1,011 bp) and have no homology.

Although we found no relationship between the ECUMN genes and the symptomatology, we cannot rule out that the proteins encoded by these genes, specifically ECUMN_0232, can target host cells, but this possibility must be proven experimentally. Since there was no relationship between gene presence and symptomatology, we analyzed the expression of ECUMN genes in the isolates containing all three genes. Interestingly, all the isolates expressed all the genes when grown in LB broth; however, when grown in urine, not all the isolates expressed all the genes. Hence, a more detailed and quantitative analysis is needed to determine whether there is a difference in expression under both conditions. We expected that all the isolates would express all the genes under both conditions, such as UPEC UMN026, because their organization suggests that they are transcribed as an operon along with vgrG (Fig. 8A) and probably hcp, as has been shown for other effector modules (Alteri et al., 2017; Miyata et al., 2013). The differences in expression observed in urine (the expression of ECUMN_0233 in all the isolates, but not that of ECUMN_0231 or vgrG) could be due to dual regulation, in which the expression of the immunity gene is driven by a second promoter located in the coding region of the effector. The presence of a second promoter ensures that the immunity gene is expressed under conditions in which the T6SS is not, thus maintaining the bacterium protected, as described for immunity proteins TsiV1, TsiV2, and TsiV3 of Vibrio cholerae (Miyata et al., 2013). In our case, this dual regulation could also apply for ECUMN_0232 since it is expressed in all isolates, but this hypothesis must be corroborated experimentally. An in silico analysis using the BDGP: neural network promoter prediction tool (https://www.fruitfly.org/seq_tools/promoter.html) (Reese, 2001) of the regions upstream of ECUMN_0232 and ECUMN_0233 (200 nt) predicted two transcriptional start sites in the coding region of ECUMN_0231 and one promoter in the coding region of ECUMN_0232 with a score >0.9. Another analysis using the iPromoter-2L tool (http://bliulab.net/iPromoter-2L/) (Liu et al., 2018) predicted sigma-70 and sigma-24 promoter regions in the coding region of ECUMN_0231, as well as sigma-70 promoter regions in ECUMN_0232. In any case a conserved sigma-70 or sigma-24 promoter with the canonic spacer regions could not be identified, so a further experimental analysis is necessary to corroborate these predictions. The expression of these UPEC genes under laboratory conditions (LB broth) differs from that of the T6SS of other bacteria, which has been shown to be repressed under these conditions (Aubert, Flannagan & Valvano, 2008; Mougous et al., 2006; Pukatzki et al., 2006; Schell et al., 2007). The fact that these genes are expressed both in LB and urine, under static and shaking conditions, suggests that they might not be involved only in pathogenesis but may also favor bacterial fitness when the bacteria are found outside the host.

UPEC UMN026 has only one T6SS, which cluster encodes the ECUMN genes. Since this SS is involved in bacterial competence, we analyzed whether this strain had antibacterial activity. Our results showed that UPEC UMN026 can kill UPEC clinical isolate 4014, which does not contain the ECUMN genes nor the T6SS. However, this strain had no activity against the clinical isolate 4012, which has the ECUMN genes and the T6SS, and expresses the ECUMN_0233 gene in both LB and urine. These results support our hypothesis that the ECUMN genes are involved in bacterial competence, probably because they constitute a chaperone/effector/immunity system, since strain UMN026 can lyse a clinical isolate that lacks these genes, but the clinical isolate carrying these genes is immune to the attack. UPEC UMN026 also showed antibacterial activity against E. faecalis. This microorganism is a Gram-positive bacterium that is considered part of the urinary microbiota, but along with E. faecium, it is the second most common cause of complicated UTIs and is responsible for 7–25% of these infections, only after UPEC infections (Alvarez-Artero et al., 2021). Our results suggest that UPEC UMN026 can compete with other UPEC strains as well as with E. faecalis inside and outside the host.

The inter- and intrabacterial activities shown by UPEC UMN026 are probably due to the ECUMN genes. Our bioinformatic analysis revealed that ECUMN genes encode a putative chaperone/effector/immunity system. ECUMN_0231 could be the chaperone since it possesses a DUF4123 domain. Proteins containing this domain have been named TEC (T6S effector chaperone) or TAP (T6 adaptor protein) and are necessary for the secretion of effectors that lack a PAAR domain (Liang et al., 2015; Unterweger et al., 2015). Other proteins with similar function contain a DUF1795 domain and are needed for the secretion of PAAR domain-containing effectors (Alcoforado Diniz & Coulthurst, 2015; Cianfanelli et al., 2016; Whitney et al., 2015). Consistent with this, we found no PAAR or other known motifs in ECUMN_0232, which encodes a putative effector. Effectors secreted by the T6SS that act within the periplasm do not contain a signal peptide to avoid self-intoxication (Wood et al., 2019). Since ECUMN_0232 does not have a signal peptide, it may target the periplasm of prey bacteria. As mentioned before, ECUMN_0233 has a signal peptide. This motif, which targets proteins to the periplasm, has been found in immunity proteins of phospholipase effectors (Ringel, Hu & Basler, 2017; Wood et al., 2019). Thus, ECUMN_0233 encodes a putative immunity protein.

Docking analysis revealed that ECUMN_0231 interacts with ECUMN_0232, as expected for a TEC and its cognate effector, and that ECUMN_0232 interacts with ECUMN_0233, as expected for an effector and its cognate immunity protein. Interestingly, the residues of ECUMN_0231 involved in the interaction with ECUMN_0232 are not located in the DUF4123 domain, so further experimental analysis to determine the function of this domain is necessary. ECUMN_0232 interacts with ECUMN_0231 via its N-terminal end, and with ECUMN_0233 via its C-terminal end. This could suggest that the active site of ECUMN_0232 is located on its C-terminal end since immunity proteins inactivate their cognate effectors by binding to the active site (Benz et al., 2012; Hagan et al., 2023; Robb et al., 2016).

In conclusion, the ECUMN genes are conserved in clinical isolates from symptomatic and asymptomatic patients and are not related to UTI symptoms. However, these genes encode a putative chaperone/effector/immunity system that seems to be involved in the antibacterial activity of strain UMN026.

Supplemental Information

Supplemental Information 1 Database of UPEC and ECUMN genes.

Supplemental Information 2 Uncropped gels of Figures 2-6.

Supplemental Information 3 Raw data for antibacterial activity.

These data were used for statistical analysis to determine the relation between ECUMN and vgrG genes and symptomatology.

Supplemental Information 4 Representative images of competition assays.

The bacterial growth at different dilutions (10-1 - 10-6) of the prey strains (A) UPEC 4014 and (B) E. faecalis after the interaction with attacking strains UPEC UMN026 and the negative control UPEC 4018 in LB and urine.

Additional Information and Declarations

Competing Interests

Author Contributions

Data Availability

The authors declare that they have no competing interests.

América Casiano González performed the experiments, analyzed the data, prepared figures and/or tables, and approved the final draft.

Arantxa Pacheco Villanueva performed the experiments, analyzed the data, prepared figures and/or tables, and approved the final draft.

Natividad Castro-Alarcón conceived and designed the experiments, analyzed the data, authored or reviewed drafts of the article, and approved the final draft.

Julio Méndez performed the experiments, analyzed the data, prepared figures and/or tables, and approved the final draft.

Ricardo Oropeza conceived and designed the experiments, analyzed the data, authored or reviewed drafts of the article, and approved the final draft.

Verónica I. Martínez-Santos conceived and designed the experiments, analyzed the data, prepared figures and/or tables, authored or reviewed drafts of the article, and approved the final draft.

The following information was supplied regarding data availability:

The raw data of the presence of the ECUMN and vgrG genes and patient symptomatology are available in the Supplemental File.

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
