# Peer review of "A novel chaperone-effector-immunity system identified in uropathogenic Escherichia coli UMN026"

_PeerJ, doi:10.7717/peerj.17336_

## Round 0.1 · original submission · Major Revisions

Three experts in this field assessed your manuscript and raised several concerns that need to be addressed before reaching a final decision. Among them, the most relevant ones are the inclusion of a more detailed methodology, the expansion of the bioinformatics analyses, the improvement of the discussion, and the quality of the English used to prepare this manuscript

**Language Note:** The review process has identified that the English language must be improved. PeerJ can provide language editing services - please contact us at [email protected] for pricing (be sure to provide your manuscript number and title). Alternatively, you should make your own arrangements to improve the language quality and provide details in your response letter. – PeerJ Staff

·

Basic reporting

The work by Casiano-Gonzalez and colleagues surveyed 99 clinical isolates for the frequency of vgrG genes in UTI isolates. It established the relationship between these genes and the phenotype of the strains tested.
Overall, the manuscript is well and professionally written, and the results presented are relevant. In the following sections, I provide some comments and concerns regarding the manuscript, which are in good faith and hopefully will help improve the manuscript.

Experimental design

Overall, the methods section has enough detail to support the findings. However, information missing needs to be added to the manuscript.
Please add a reference for the DNA extraction method in the methods section. Also, I suggest fusing the RNA purification section with RT reactions.
Although the statistical test and values are indicated in the tables and figures, and the analysis is correct, authors must include in the methods section a statistics section showing the statistical tests used, the software to conduct the statistical analysis, and the number of replicates (either in the figure legend and tables or in the methods section. For this reviewer, this is mandatory. Also, please add the methods used for the bioinformatic analysis in the methods section to ensure reproducibility and clarity.
Please include in Table 1 the size of the amplicon and Tm values for the primers used.
Finally, an explanation is needed as to why the authors did not use qRT-PCR to analyze the expression patterns of the genes of interest. Please add this in the rebuttal letter.

Validity of the findings

Accompanying the introduction, I recommend including a diagram of the gene structure of T6SS in the different gene clusters, and this can be Fig. 1. The paper is relevant and contains important results, so I recommend including information that will be helpful to non-specialist readers.
In lines 186 and 187, the authors used the expression “more intense band.”, since the authors did not use qRT-PCR or an endogenous housekeeping gene as an internal control to show that the expression is constant, this expression is invalid. In this reviewer's opinion, the authors aim to show that genes are expressed, which they successfully show. Thus, I recommend toning down the manuscript to “expression was found” and not to try to extrapolate to “more or less” expression. I think the qualitative expression shown here is sufficient for the manuscript. So, when referring to Fig. 3 to 5 I suggest limiting the interpretation to “is expressed” or “is not expressed”. The explanation in the discussion (using the same amount of cDNA) is not satisfactory and I suggest removing this statement. Without proper expression controls (endogenous and multiplex) or the use of qRT-PCR, the authors should limit the paper to whether is expressed or not.
To further support the results shown in Fig. 7, I kindly request to show the antibacterial effect on the sensitive strains such as 4014 and E. faecalis as supplementary material.
This reviewer finds the bioinformatic analysis limited to the available tools, and the authors can provide stronger evidence for the chaperon homology. UniProt's available crystal structures and protein models can provide stronger evidence for the chaperon homology the authors found. I suggest adding a structural model comparison and identifying the key residues involved in the chaperon activity by sequence alignment. Signal peptide identification should be provided with at least two different tools, and in the methods section, indicate the tools used and the residues spanning the signal peptide along with the key residues the Sec system recognizes. The map in Figure A should also contain the size of the genes indicated or at least a reference size bar.
In the discussion regarding the expression levels found in urine, I suggest including a bioinformatic analysis to pinpoint internal promoter sequences or alternative transcription start points in the genes of these operons. The analysis could ultimately support the findings of differential regulation in these genes.
Overall, the strength of the manuscript is good. Still, this reviewer thinks that the lack of bioinformatic analysis that unambiguously demonstrates that the effector protein is indeed a chaperon of the T6SS opens other possibilities, as Vasallo and colleagues (doi: 10.1038/s41564-022-01219-4) reported that proteins with unknown function were identified as novel anti-phage defense systems related to the toxin-antitoxin systems.

Additional comments

In line 60, please add a reference, as a suggestion, to a review encompassing the overall virulence factors for UPEC. This suggestion is to ensure that specialists can follow the work and references for non-specialists; this ensures your paper will be cited.
For clarity, I suggest deleting in line 80 “Due to this”.
I suggest in line 84 adding “…of the vgrG gene (a core component of the TGSS); to…”
In line 120 a space is missing in “in1%”; please correct.
In line 152, I recommend changing to “As stated in the introduction, …”
In line 154, please change to “In order to corroborate the genotype, we…” This reviewer thinks that phrases with unclear antecedents are hard to follow.
In line 241, please simplify to “…isolates from symptomatic as well as in asymptomatic patients”.
In line 243, please add “can be explained because.”
Please revise the writing of lines 248-249, for this reviewer is a bit hard to follow.
In line 253 please add et al.,
In line 294 please change to lyse instead of lysate.
This reviewer has final concern, the quality of the figures showing agarose gels is not ideal, specially for gels in Figure 2 panels A, C, E and G; Figure 3 panel B; Figure 4 panel A; Figure 5 panels A, B (to faint) and Figure 6 panel A. For this reviewer is not critical but if they can be improved that will provide a better impression of the manuscript.

Reviewer 2 ·

Basic reporting

The English language should be improved to ensure that an international audience can clearly understand your text. Some examples where the language could be improved include lines 46, 277, 278 – the current phrasing makes comprehension difficult. I suggest you have a colleague who is proficient in English and familiar with the subject matter review your manuscript, or contact a professional editing service.
For example, the term lysate has been used by the authors at several instances (lines 197, 200, 294) in the manuscript. However, switching it to other terms like lyse/ promote lysis/ bacteriolytic would have been more appropriate.
The authors have studied the frequency of VgrG gene occurrence in the isolates. A brief background on VgrG genes and their significance would help the readers.
Similarly, it will be beneficial to introduce effector-immunity gene pairs in better detail. Perhaps, with examples from the literature.

Experimental design

The authors have performed PCR based assay to determine the relationship between the three ECUMN_0231, ECUMN_0232, and ECUMN_0233 genes and UTI symptomology. Their experimental results have shown that the ECUMN gene occurrence and urinary tract infection could not be corelated. Furthermore, the authors wanted to show the actual function of ECUMN genes in the Type 6 secretion system. To do this, they have opted bioinformatics/ in-silico approach and claimed that the ECUMN genes encode a putative chaperone/ effector/ immunity system. However, it is unclear on how the authors came to this conclusion as both the methods and result section lacks the details of the bioinformatics tool or protocol.

Validity of the findings

For the first part of the study, the authors have used a PCR based approach to study the prevalence of ECUMN genes in UTI isolates. Also, they have performed bacterial competitive assay to show the bacterolytic property of UPEC UMN026 towards E. faecalis and other pathogenic bacteria. The methodology and data presented overall agrees well.

However, in the final part of the study, the authors have not provided sufficient information on how the concluded the three ECUMN genes performs a chaperone/ effector/ immunity function. The authors can explain their in-silico work in better details. For example, is there any online tool that has been used for domain identification?, are their examples of other proteins with similar domain and said function?

For better readability and validation, some of my suggestions to the authors for the bioinformatics portion.
1. Sequence similarity (either DNA/ protein level)- BLASTN/ BLASTP
2. Showing examples of characteristic protein domain from literature and comparing them to ECUMN genes
3. Structure similarity analysis
4. Experimentally characterizing ECUMN gene products to prove the said functionality or literature examples.

Reviewer 3 ·

Basic reporting

1. Abstract need to be improvised and rewritten.
2. Page 1 line 29. LB medium is not abbreviated before in the paragraph.
3. Grammatical errors need to be rectified.

Experimental design

The manuscript enitited “ A novel chaperone-effector-immunity system identified in uropathogenic Escherichia coli UMN026” by América Casiano-González et a., describes the analysis of ECUMN_0231, ECUMN_0232, ECUMN_0233, and vgrG genes in UTI isolates and their expression in LB medium and urine. The present work also highlights the functions of these genes in UTI symptoms or bacterial competence.

Comments:
1. Abstract need to be improvised and rewritten.
2. Page 1 line 29. LB medium is not abbreviated before in the paragraph.
3. Grammatical errors need to be rectified.
4. The conclusion part is not clear and not done based on the current research proposal.

Validity of the findings

Novelty is less and more supportive evidence or studies needed to be performed.

---

## Round 0.2 · accepted · Accept

The authors improved the manuscript, taking into account all the concerns previously raised by Reviewers. As a consequence, it is now suitable for publication.

·

Basic reporting

This reviewer thanks the authors for addressing all the major points suggested. The bioinformatic analysis added is sufficient. I think the manuscript is now suitable for publication.

Experimental design

The experimental design is in agreement with the research question and no further comment on this section is needed.

Validity of the findings

With the additional findings the paper is robust and addresses all the questions and suggestions done by this reviewer.

Additional comments

Overall, is a solid study and the paper is now suitable for publication. Good luck with your research moving forward.

Reviewer 2 ·

Basic reporting

no comment

Experimental design

no comment

Validity of the findings

The authors have satisfactorily addressed the concerns that was put forth in the review process. The authors have also explained the bioinformatic methodology used for protein-protein interaction prediction.